# Poor Long-Term Renal Allograft Survival in Patients with Chronic Antibody-Mediated Rejection, Irrespective of Treatment—A Single Center Retrospective Study

**DOI:** 10.3390/jcm11010199

**Published:** 2021-12-30

**Authors:** Kaiyin Wu, Danilo Schmidt, Covadonga López del Moral, Bilgin Osmanodja, Nils Lachmann, Qiang Zhang, Fabian Halleck, Mira Choi, Friederike Bachmann, Simon Ronicke, Wiebke Duettmann, Marcel G. Naik, Eva Schrezenmeier, Birgit Rudolph, Klemens Budde

**Affiliations:** 1Department of Nephrology and Medical Intensive Care, Charité Universitätsmedizin Berlin, 10117 Berlin, Germany; danilo.schmidt@charite.de (D.S.); covadonga.lopez@charite.de (C.L.d.M.); bilgin.osmanodja@charite.de (B.O.); qiang.zhang@charite.de (Q.Z.); fabian.halleck@charite.de (F.H.); mira.choi@charite.de (M.C.); friederike.bachmann@charite.de (F.B.); simon.ronicke@charite.de (S.R.); wiebke.duettmann@charite.de (W.D.); marcel.naik@charite.de (M.G.N.); eva-vanessa.schrezenmeier@charite.de (E.S.); klemens.budde@charite.de (K.B.); 2Institute of Transfusion Medicine, Charité Universitätsmedizin Berlin, 10117 Berlin, Germany; nils.lachmann@charite.de; 3Berlin Institute of Health, Anna-Louisa-Karsch-Str. 2, 10178 Berlin, Germany; 4Institute of Pathology, Charité Universitätsmedizin Berlin, 10117 Berlin, Germany; birgit.rudolph@charite.de

**Keywords:** kidney transplantation, chronic antibody-mediated rejection, transplant glomerulopathy, antihumoral therapy, graft survival

## Abstract

The Banff 2017 report permits the diagnosis of pure chronic antibody-mediated rejection (cAMR) in absence of microcirculation inflammation. We retrospectively investigated renal allograft function and long-term outcomes of 67 patients with cAMR, and compared patients who received antihumoral therapy (cAMR-AHT, *n* = 21) with patients without treatment (cAMRwo, *n* = 46). At baseline, the cAMR-AHT group had more concomitant T-cell-mediated rejection (9/46 (19.2%) vs. 10/21 (47.6%); *p* = 0.04), a higher g-lesion score (0.4 ± 0.5 versus 0.1 ± 0.3; *p* = 0.01) and a higher median eGFR decline in the six months prior to biopsy (6.6 vs. 3.0 mL/min; *p* = 0.04). The median eGFR decline six months after biopsy was comparable (2.6 vs. 4.9 mL/min, *p* = 0.61) between both groups, and three-year graft survival after biopsy was statistically lower in the cAMR-AHT group (35.0% vs. 61.0%, *p* = 0.03). Patients who received AHT had more infections (0.38 vs. 0.20 infections/patient; *p* = 0.04). Currently, antihumoral therapy is more often administered to patients with cAMR and rapidly deteriorating renal function or concomitant TCMR. However, long-term graft outcomes remain poor, despite treatment.

## 1. Introduction

In the last decade, it became evident that chronic antibody-mediated injury is an important cause of late allograft failure, outside of death, with functioning graft [1]. The transplant glomerulopathy (TG), also known as Banff cg-lesion, with characteristic duplication and/or multilayering of the glomerular basement membrane, represents the morphological substrate of chronic antibody-mediated injury in the kidney allograft [2,3,4]. With the concurrent improvements in HLA-antibody detection methods, important advances have been made in the diagnosis and pathophysiology of AMR, and a wide spectrum of subcategories have been introduced into the Banff classification, which comprise the phenotypic heterogeneity of AMR [5]. The chronic active AMR (cAAMR) was first defined in Banff 2005 report [6]; the diagnosis of cAAMR required coexistence of the morphologic features (in most cases of TG), the immunohistologic evidence (capillary C4d deposition) and the serologic evidence for circulating donor specific HLA-antibodies (DSA). Later, a C4d-negative cAAMR was recognized in Banff 2013 report [7], and peritubular capillary C4d deposits could be replaced by at least moderate microcirculation inflammation (MVI). Nevertheless, in the clinical setting, it is common for the three diagnostic features of cAAMR to appear as an incomplete combination. As a consequence, the Banff 2017 report [8] permits the diagnosis of pure cAMR based on the presence of TG and circulating DSA in the absence of capillary C4d deposits or MVI (MVI score ≤ 1). However, the repeatedly revised criteria of AMR leaves clinicians uncertain about the outcome after diagnosis and the consequences for treatment. 

So far much effort has been made to investigate the therapeutic options for attenuating the chronic antibody-mediated destruction [9]. The primary aims of nearly all therapeutic approaches for AMR are removing circulating DSA and reducing DSA production. In this sense, the strongholds for contemporary treatment of active AMR are represented by high-dose intravenous immunoglobulin (IVIG) and plasmapheresis (PPh); it has been demonstrated that IVIG plus PPh, with or without rituximab, a frequently prescribed standard of care for active AMR, could improve short-term outcomes, while their results on long-term effects remain variable [10]. Moreover, there is currently no established consensus for the treatment of pure chronic AMR without microcirculation inflammation. The treatment studies for chronic active AMR are rarely comparable, partly because different Banff classifications were used, and the available evidence is generally of low quality. Several studies have reported that the presence of TG associates with a low response to therapy and, consequently, constitutes one of the main reasons for the late graft failure [11,12]. Due to the low evidence for the outcome of patients with pure cAMR according to Banff 2017, we performed a single-center retrospective study to investigate long-term outcomes with a focus on the efficacy and safety of currently available agents for treatment of pure cAMR. 

## 2. Materials and Methods

### 2.1. Study Population and Data Collection

We reviewed all adult recipients who received the single kidney transplantation in the transplant centre of Charité Campus Mitte and Charité Virchow Klinikum. Between January 2008 and December 2018, 4380 for-cause graft biopsies were performed, and TG was observed in 487 biopsies (11.1%) in 325 patients. All patients who developed pure cAMR with TG in kidney grafts were enrolled in this retrospective study. The exclusion schema involved an active malignant disease, recurrent or de novo glomerulonephritis or de novo thrombotic microangiopathy (TMA) in allograft and polyoma virus nephropathy (Figure 1). All renal allograft recipients with cAMR according to Banff 2017 and who routinely visited the transplantation clinic for follow-up care were enrolled. The demographic, transplantation characteristics, immunosuppression and treatment were registered at each outpatient clinic visit in the database [13], and the measurements of eGFR, serum creatinine and proteinuria were taken six months before and at studied biopsy as well as every three months after diagnosis. The database was almost complete, with <6% missing values in different data fields. In the case of missing values at a certain time point, the next available value was used. If there were several measurements in one time interval, the measurements at or nearest to the planned follow-up were used for analysis. In addition, measurements taken during hospitalization were omitted from analysis to minimize bias due to intercurrent illness and treatment, for example, infection and the admission of intravenous fluids etc. All clinical and laboratory data were selected in the transplant database system [13] and assessed for completeness by a single investigator (S.D.). 

### 2.2. Histopathology Evaluation

All for-cause biopsies were undertaken when the serum creatinine (Scr) rose above 25% from the baseline or there was a clinical relevant increase in daily urinary protein excretion. The biopsy specimens were processed with standard techniques in the institute of pathology. All histological slides of the studied biopsy were selected from the archive and reevaluated by the nephropathologist (B.R.) according to the Banff classification 2017 [8]. TG was distinguished from recurrent or de novo immune complex glomerulopathy by the immunofluorescence and electron microscopy [14]. The C4d deposition was detected by indirect immunofluorescence on paraffin sections of formalin-fixed tissue (polyclonal anti-C4d antibody, Dianovo, Germany); more than 10% peritubular capillaries with linear deposition of C4d were considered as a positive reaction. In addition, the proximal tubular epithelial cytoplasma vacuolization and obliterative arteriopathy were often referred to chronic calcineurin-inhibitor (CNI) toxicity. All Banff lesions were graded on a scale of 0–3 according to the proportion of cortical area affected, with higher scores indicating more severe abnormalities. 

### 2.3. HLA-Antibodies Screening

All serum samples were collected post studied biopsy and qualitatively screened for HLA antibodies (HLA−ab) by the Luminex assay [15] from 2007 on. In addition, HLA-ab specificities were determined by LABScreen Single Antigen beads assay (One Lambda, Canoga Park, CA, USA). As an indicator for the antibody level, the normalized fluorescent intensity (MFI) of the immunodominant donor-specific antibody was used. HLA-ab were considered positive when exceeding a plausible MFI value of >500. The values of MFI of immunodominant DSA against class I (panel A) or class II (panel B) antigens were examined at biopsy and during the first year after studied biopsy. All tests were performed according to the manufacturer’s guidelines, and the DSA level were monitored in regular intervals, as previously described [16]. 

### 2.4. Immunosuppression and Therapeutic Strategies

The maintenance immunosuppressive protocols are shown in Table 1. The doses of cyclosporine A (CyA)/tacrolimus (Tac) were adjusted according to whole blood trough levels. For treatment of AMR, PPh was performed for six sessions, and IVIG at 1.5 g/kg were given [17]. In addition, 5 of 21 patients received a single dose of rituximab (375 mg/m^2^ body surface area) one week after the last IVIG infusion, and 4 patients received bortezomib at 1.3 mg/m^2^ administered intravenously on days 1, 4, 8 and 11 after the first IVIG infusion. When the biopsy showed mixed TCMR, the patients were given 500 mg methylprednisolon for three days, and thereafter reduced to maintenance dose of 5 mg/d. After intervention, the patients received trimethoprim-sulfamethoxazole as prophylaxis for pneumocystis jirovecii for six months. When a progressive calcineurin (CNI) nephrotoxicity was observed by biopsies, the doses of CyA/Tac were minimized or switched to a CNI-free treatment of sirolimus/everolimus or belatacept [18]. 

In addition, the patients with hypertension received at least one antihypertensive drug; the patients having daily urinary protein excretion of more than 1 g/L received the maximum tolerable dose of an angiotensin converting enzyme inhibitor (ACEi) and/or an angiotensin receptor blocker (ARB). 

### 2.5. The Observation of Clinical Outcomes

Our study ended at 60 months after studied biopsy or initiation of chronic dialysis. The estimated glomerular filtration rate (eGFR milliliter per minute per 1.73 m^2^) was measured by calculated creatinine clearance equation [19]. The eGFR value of a failed graft was assigned at 5 mL/min/1.73 m^2^. Furthermore, the responses to AHT were evaluated through comparing the variation in eGFR and the daily urinary protein excretion with the period before and after studied biopsy between the cAMR-AHT and cAMRwo group. The long-term clinical outcomes were observed by comparing the graft and patient survival rates between two groups at 12, 24, 36, 48 and 60 months post studied biopsy. 

### 2.6. Statistical Analysis

Normally distributed data were expressed as mean ± SD and non-normally distributed data as median (IQR). Categorical variables were expressed as N and percentage of the total. Student’s t-test was used to compare two groups of continuous variables, and chi-square was used for categorical data. Patient- and death-censored graft survival rates were analyzed by Kaplan–Meier curves and log rank test. For univariate analysis of the histological factors influencing the five-year death-censored graft failure, we performed a Kaplan–Meier analysis for each histological Banff lesion comparing mild (score 0–1) and severe (score 2–3) lesion scores. The Log Rank test was used for statistical comparison between cases with mild and severe grade of each Banff lesion, and the Banff lesions with *p*-values < 0.05 were selected for further multivariable analysis. For multivariable modeling, a binary-logistic regression analysis was employed to examine the effects of three selected clinical factors (receiving antihumoral therapy, eGFR and proteinuria at biopsy) on overall graft survival, patient survival and death-censored graft survival. Proteinuria and eGFR were classified as dichotomous values above or below the median. Adjusted estimates from multivariable models are presented as odds ratios (OR) with 95% confidence intervals (CI). All statistics were performed by using SPSS16.0 (SPSS Inc., Chicago, IL, USA); *p*-value < 0.05 was considered as significant.

## 3. Results

### 3.1. Clinical Characteristics at Studied Biopsy

We identified 67 renal allograft recipients with pure cAMR in our database and compared long-term outcomes of patients who received antihumoral therapy (cAMR-AHT, *n* = 21) with patients without treatment (cAMRwo, *n* = 46). Most baseline demographic characteristics of patients were comparable between two cAMR groups (Table 1). The indications of allograft biopsy were also similarly distributed in two groups. However, the cAMR-AHT group had more concomitant T-cell-mediated rejection compared to cAMRwo (9/46 (19.2%) vs. 10/21 (47.6%); *p* = 0.04). TG was first observed at a median time of 7.3 years after kidney transplantation of the cAMR-AHT group, in comparison with 5.3 years for the cAMRwo group (*p* = 0.03). The serologic DSA was detectable at a median time of 5.3 years post transplantation in the cAMR-AHT group and 6.6 years in the cAMRwo group (*p* = 0.30). In addition, eight (17.4%) patients of the cAMRwo group and three (14.3%) of the cAMR-AHT group had single class I HLA-antibodies, twenty-eight (60.8%) patients of the cAMRwo group and thirteen (61.9%) of the cAMR-AHT group had class II HLA-antibodies, and ten (21.7%) patients of the cAMRwo group and five (23.8%) of the cAMR-AHT group had both class I and II HLA-antibodies. 

After biopsy, the maintenance immunosuppressive regimen remained unmodified in twelve (26.1%) patients of the cAMRwo group and four (19.0%) of the cAMR-AHT group; ten (21.7%) patients of the cAMRwo group and five (23.8%) of the cAMR-AHT group received an increased dose of CNI; nine (19.6%) patients of the cAMRwo group and three (14.3%) of the cAMR-AHT group reduced the dose of CNI; five (10.9%) patients of the cAMRwo group and one (4.8%) of the cAMR-AHT group switched from CNI to Belatacept (each pair-wise comparison yields *p* > 0.05). 

### 3.2. Effect of Antihumoral Therapy (AHT) on DSA

As shown in Table 2, no significant differences were found on the median immunofluorescence intensity (MFI) of the DSA between cAMR-AHT and cAMRwo groups at 0 days, 180 days and one year post studied biopsy.

### 3.3. Effect of AHT on the Allograft Function

As illustrated in Table 2, in the cAMR-AHT group, the median eGFR of cAMR-AHT declined from 40.1 mL/min/1.73 m^2^ at six months prior to the studied biopsy to 26.4 mL/min/1.73 m^2^ at biopsies, and further declined to 21.6 mL/min/1.73 m^2^ at six months after biopsy. The median eGFR of the cAMRwo group declined from 35.3 mL/min/1.73 m^2^ at six months before the studied biopsy to 28.5 mL/min/1.73 m^2^ at biopsy, and further declined to 26.7 mL/min/1.73 m^2^ at six months after biopsy. In the six months prior to studied biopsy, the decline of median eGFR in cAMRwo group was significantly lower than that of the cAMR-AHT group (6.6 (−6.0–30.0) vs. 13.1 (−1.0–60.6) mL/min/1.73 m^2^; *p* = 0.04). The median eGFR decline six months after biopsy was similar between groups (*p* > 0.05). 

Interestingly, twenty (21.7%) patients in the cAMRwo group and seven (33.3%) patients in cAMR-AHT group had more than 1000 mg/d proteinuria at indication biopsy (Table 2). The median of daily proteinuria at six months pre-, at- and six months post-biopsy were comparable between the two groups (each pair-wise comparison yields *p* > 0.05). In addition, 40 (85.7%) patients of the cAMRwo group and 19 (90.5%) of the cAMR-AHT group received antihypertensive therapy with at least one ACE inhibitor or ARBs (*p* = 0.80). 

### 3.4. Effect of AHT on the Long-Term Clinical Outcomes

The five-year Kaplan–Meier estimate for DCGS after diagnosis of cAMR was 32.7%. As illustrated in Figure 2, the two- and three-year DCGS rate of the cAMR-AHT group was significantly lower than those of the cAMRwo group (46.7% vs. 76.2% at two-year, *p* = 0.01; 35.0% vs. 61.0% at three-year, *p* = 0.02). At one, four and five years post biopsy, the DCGS rates of the cAMR-AHT group were lower than those of the cAMRwo group, though the differences did not reach significance, with low numbers. 

Patient survival rates were similar between the two groups at each time up to five years post studied biopsy. Finally, the five-year Kaplan–Meier estimate for overall graft survival (including patient death) was 31.0% (cAMR-AHT group 35.9% vs. cAMRwo group 21.1%, *p* = 0.06).

### 3.5. Histological Evaluation of the Studied Biopsy

The mean scores of the Banff lesions at biopsy were shown in Table 3; the mean g-lesion score of the cAMR-AHT group was significantly higher than that of the cAMRwo group (0.4 ± 0.5 versus 0.1 ± 0.3; *p* = 0.01). As mentioned, concomitant TCMR was observed more frequently in the cAMR-AHT group (*p* = 0.04). In addition, the cAMRwo group showed statistically higher mean scores of ci- and ct-lesion in comparison to the cAMR-AHT group (each pair-wise comparison yields *p* < 0.05).

Twenty (42.6%) patients of the cAMRwo group and eleven (52.4%) of the cAMR-AHT group experienced at least one for-cause biopsy during the follow-up after the studied biopsy (*p* = 0.76), of which nine (19.1%) patients of the cAMRwo group and four (19.0%) of the cAMR-AHT group developed cAAMR (*p* = 0.85).

### 3.6. Safety of AHT

During the six months after biopsy, the infection complications (ICs) that required hospitalization occurred more frequently in the cAMR-AHT group than in the cAMRwo group, with a ratio of ICs/patient of 0.38 vs. 0.19 (*p* = 0.04, Table 1). Two (9.5%) cAMR-AHT patients experienced CMV infectious colitis, in comparison to none of the cAMRwo group (*p* = 0.02).

### 3.7. Correlation of Histological and Clinical Features with Five-Year Outcome after Diagnosis of cAMR

Each Banff lesion was divided into mild grade (score 0–1) and severe grade (score 2–3). After the exclusion of three patients, who died with a functioning graft, we could not find significant differences in five-year death-censored graft survival when comparing mild and severe Banff lesion scores in univariate Kaplan–Meier analysis (Appendix A). As a consequence, no histological Banff lesion was selected for further multivariable analysis. Based on clinical experience, we performed a binary-logistic regression to assess the association of three selected clinical variables (eGFR (above or below median), proteinuria (above or below median) and receiving antihumoral therapy) with five-year post-biopsy overall graft survival, patient survival and death-censored graft survival; no histological or clinical factors were found to be significantly associated with the long-term outcome (Table 4).

## 4. Discussion

With the increasing recognition of the role of alloantibodies and the corresponding morphological changes in patients with chronic deterioration of allograft function, the diagnostic criteria for AMR have been considerably updated [20]. However, there is still significant ambiguity in the Banff criteria in terms of the classification and behavior of the different forms of AMR [11,21]. The current criteria of cAMR was last updated in the Banff 2017 report; unlike cAAMR, the capillary C4d deposition and at least moderate MVI are no longer conditions for the diagnosis of cAMR [8]. Nevertheless, there is still no proven treatment to modify the natural course of cAMR, and the adverse events derived from these AHT strategies are of great concern [22,23]. In the present study, we evaluated whether a therapeutic regimen consisting of IVIG/PPh and other drugs frequently used for treatment of active AMR could delay the rate of renal function decline in pure cAMR patients. To select a homogeneous population, we excluded any biopsies showing TG with C4d positive staining or at least moderate MVI to minimize any potential treatment effect for patients with cAAMR. Our retrospective analysis did not show any obvious benefit of AHT in patients with cAMR with regard to long-term outcomes. While we cannot exclude a potential positive effect on the reduction in the observed rapid eGFR decline, we noted a relevant overimmunosuppression after AHT. 

Nearly all therapeutic approaches for treating AMR aim to remove circulating DSA and to reduce DSA production [24]. A high dose of IVIG remains an essential component of AHT, and it was suggested that better outcomes are achieved when IVIG is accompanied with PPh for depleting circulating DSAs [25]. Although IVIG/PPh is regarded as the ”standard care of AMR“ [26], it exerts, in most cases, only on the effect of turning an acute form of AMR into a subclinical disease, since the DSA-producing plasma cells are not affected [27]. In an attempt to prevent further antibody production, some patients received additional rituximab or bortezomib therapy. A prospective, double-blind, multicenter, randomized study [28] reported that treatment of late AMR with rituximab, in combination with steroids, IVIG and PPh, did not improve any outcome parameter compared to placebo. Only side effects were increased, with a higher risk of infection [29]. An observational cohort study suggested that the therapeutic efficacy of bortezomib in combination with PPh and IVIG is higher in early AMR, while patients with late AMR were less responsive, suggesting limited treatment efficacy in this indication [30]. In summary, current evidence is in line with our data and supporting the conclusion of a recent consensus conference [11] that there is no proven treatment for cAMR.

The development of TG is viewed as a structural ‘end-product’ of the antibody-mediated pathophysiological process [31], while the quality and quantity (titer) of circulating DSAs may impact the clinical manifestation of the AMR [32,33]. However, discrepancies between histological and serological findings are common. Features of active AMR, such as C4d deposition and a high grade of MVI lesions, may be fluctuating and cannot be predicted by positive serology alone. Patients with exclusively weak or no complement-activating DSAs tend to experience less disease activity and may have better outcomes [34,35]. Moreover, it is well documented that different clinical courses after development of DSA exist [36]: some patients with detected DSAs develop one or more episodes of acute graft dysfunction associated with AMR [37]; on the contrary, some patients with serologic DSAs experience only subclinical AMR [38]. In this respect, even histological discrepancy exists, and it is discussed that cAMR and cAAMR represent a spectrum of disease severity of the same disorder, instead of two distinct pathophysiological types. Our study showed that nearly 19% of cases in both groups developed the cAAMR in follow-up biopsies, providing further evidence for a fluctuating disease activity and/or patchy distribution of disease activity in the kidney. Our clinical observation may also support the potential utility of a protocol biopsy in patients with unsuccessfully treated TG in further studies. 

Transplant glomerulitis (Banff g-lesion) is one of the morphological features indicating active AMR and defined as a Banff g-lesion. Although cAMR groups in our study had maximal g1-lesion, the mean score of the g-lesion was statistically higher in the cAMR-AHT group than in the cAMRwo group. Even mild glomerulitis together with rapid deterioration of graft function led the physician to consider AHT more frequently, in the hope of modifying progression. Although we cannot exclude some potential minor positive effects, our study revealed that the infectious adverse events and hospitalization were more frequent in the cAMR-AHT group. Similar to previous reports, the AHT regimen with enhanced immunosuppression led to a higher number of overimmunosuppression—in particular, opportunistic infections [39]—as well as conferred a substantial risk of drug-toxicities, which was closely associated with the deterioration of the tubulointerstitial fibrosis and inferior late graft survival [40,41]. Furthermore, the changes in maintenance immunosuppressive therapy post studied biopsy may complicate the evaluation of the safety of the AHT regimen. Therefore, the ideal therapeutic guidelines for cAMR remain to be determined, and the choice of appropriate medication dosage, paired with careful patient monitoring and adjustment of baseline immunosuppression, needs to be considered.

AMR is often initially detected with concomitant TCMR, and the treatment of concomitant TCMR is recommended in all cases of AMR [42]. Our data showed that concomitant TCMR were detected in significantly more cases of the cAMR-AHT group than that of the cAMRwo group, in parallel with the evidently rapid decline in eGFR before studied biopsy in the cAMR-AHT group than that of the cAMRwo group. Consequently, an additional steroid bolus was given to treat the mixed TCMR. The fact that median eGFR decline was similar between the cAMR-AHT and cAMRwo groups may suggest that the concomitant TCMR responded to additional steroid bolus treatment, albeit did not improve the graft outcome in cAMR patients. 

While AHT was preferred in patients with cAMR and rapidly progressing allograft insufficiency, conservative treatment was chosen in patients with advanced interstitial fibrosis/tubular atrophy (IFTA) but minimal features of active AMR. Chronic histologic damage has previously been identified as one of the most important attributing factors to kidney allograft loss, irrespective of diagnosis [43]. We cannot discard the possibility that the lack of response to AHT may be due to the selection of a population with too advanced and irreversible pathological damage. Although the long-term outcome was poor, conservative treatment approaches were even better than AHT. Long term exposure to CNI is one of the major risk factors leading to arterial intimal fibroproliferation and neointimal thickening, eventually resulting in graft ischemia and striped tubulointerstitial fibrosis [44]. The patients in both the cAMR-AHT and cAMRwo groups displayed moderate to severe transplant vasculopathy, which are regarded as the hallmarks of chronic allograft dysfunction and undoubtedly contribute to late graft loss [45]. In addition, the morphologic feature of IFTA (Banff ci- and ct-lesions) is commonly seen in the late period of transplantation and indicates the cumulative burden of injury and diseases [46]. Patri et al. have previously shown that a chronic inflammation score combines interstitial fibrosis and tubular atrophy associated with long-term graft failure in patients with TG [47]. In our study, although the mean score of ci- and ct-lesion in both groups had not yet advanced to the chronic scarring stage, the presence of IFTA in combination with severe vasculopathy and glomerulosclerosis predicted rather poor graft survival. 

At biopsy and six months post studied biopsy the eGFR and proteinuria levels were similar between the two cAMR groups. Furthermore, the MFI of the immunodominant DSA presented similarly at studied biopsy and during follow-up. Differences in scores associated with chronic scarring might be considered secondary to the diabetes and hypertension, and might be partially explained by the evidently higher level of Hb1Ac and systolic blood pressure in the cAMRwo group. Despite lower mean scores of mm-, ci- and ct-lesions, patients receiving AHT had an inferior graft outcome. The logistic regression analysis also showed that receiving AHT had no significant impact on long-term graft outcome.

In summary, our observational study suggests that the utility of the AHT regimen to treat cAMR had no major effect on the fluctuating course of DSA or improving the graft outcomes. Even worse, the AHT regimen created infectious complications for the patients. We recognize that a prospective, randomized trial will be needed to validate these initial findings, and our findings clearly support the use of a rigorous control group. Meanwhile, our evidence suggests that, when approaching the use of existing AHT agents for inactive AMR, less may be more.

## Figures and Tables

**Figure 1 jcm-11-00199-f001:**
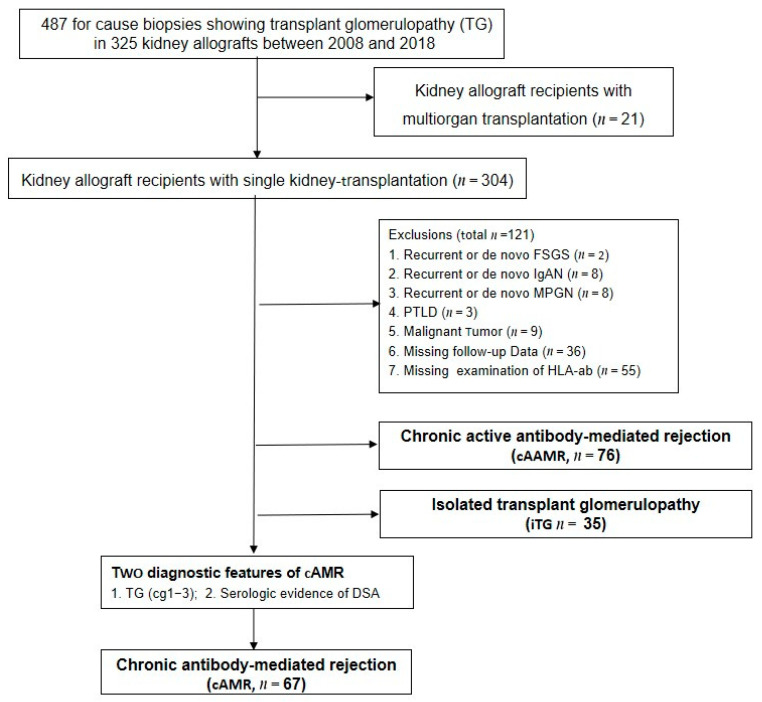
Flow chart of patients enrolled in the study. iTG: isolated transplant glomerulopathy; cAMR: chronic antibody-mediated rejection; cAAMR: chronic active antibody-mediated rejection; PTLD: post transplant lymphoproliferative disorder; IgAN: IgA Nephropathy; FSGS: focal segmental glomerulosclerosis; HLA-ab: antibody against human leukocyte antigen; TG: transplant glomerulopathy; DSA: donor specific anti-HLA antibody.

**Figure 2 jcm-11-00199-f002:**
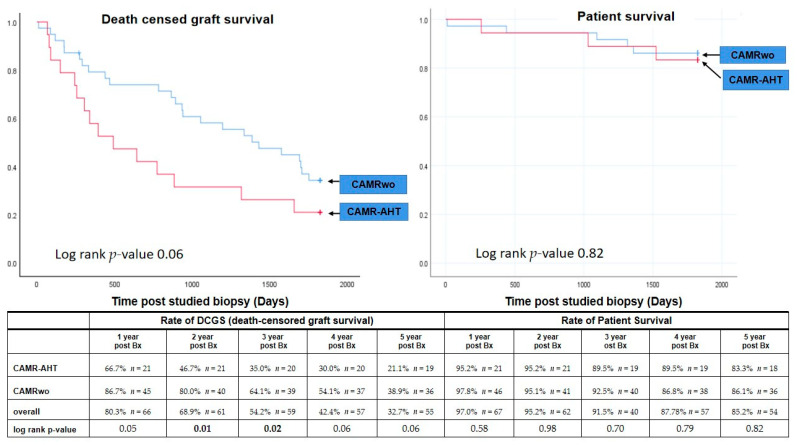
The comparison of death-censored graft survival rates and patient survival rates up to 5 years post biopsy between cAMR-AHT and cAMRwo groups.

**Table 1 jcm-11-00199-t001:** Demographics and clinical characteristics of patients with cAMR.

**(A)** **Demographics**			
	cAMRwo (*n* = 46)	cAMR-AHT (*n* = 21)	*p*-Value
Recipient age (years, median IQR)	43 (18–69)	41 (25–67)	0.99
Recipient gender (m/f)	21/25	9/12	0.41
Recipient BMI (kg/m^2^ median, IQR)	24.8 (18.3–35.5)	22.8 (19.7–34.4)	0.40
First kidney transplant N (%)	31 (67.4%)	15 (71.4%)	0.83
PRA before transplantation >10% N (%)	4 (8.7%)	5 (23.8%)	0.24
Broad HLA-mismatches (N, median IQR)	3 (0–6)	3 (0–5)	0.75
CIT (hours median IQR)	5.8 (0.5–20.3)	7.5(0.5–18.5)	**0.03**
Presence of DGF N (%)	14 (32.6%)	11 (52.4%)	0.10
Donor age (years, median, IQR)	50 (17–83)	49 (16–71)	0.21
Donor gender (m/f)	19/25	10/11	0.53
Living donation N (%)	18 (40.0%)	7 (33.3%)	0.55
**(B)** **Clinical characteristics**			
Indications of the studied biopsy for graft dysfunction/proteinuria/or both (N)	21/5/18	15/0/6	0.09
Duration between transplantation and studied biopsy (years, median IQR)	7.3 (1.2–18.7)	5.3 (0.3–14.2)	**0.03**
Follow-up after biopsy (years, median IQR)	5.6 (0.5–15.0)	6.3 (0.7–11.5)	0.10
Detectable DSA in serum (months post transplantation, median IQR)	79.1 (14.8–197.6)	63.9 (0.5–177.0)	0.90
HLA-antibody class type I/II/both	8/28/10	3/13/5	0.58
**Maintenance immunosuppression regimens at studied biopsy** N (%)			
Tac + MMF/MPA + PDN	24 (52.2%)	16 (76.2%)	0.06
CyA + MMF/MPA + PDN	6 (13.0%)	1 (4.8%)	0.29
Rapamycin + MMF/MPA + PDN	2 (4.3%)	1 (4.8%)	0.91
Tac + MMF/MPA	1 (2.2%)	1 (4.8%)	0.22
CyA + MMF/MPA	6 (13.0%)	1 (4.8%)	0.38
CyA + Azathioprine + PDN	1 (2.2%)	0 (0.0%)	0.45
Tac + PDN	3 (6.5%)	0 (0.0%)	0.40
CyA + PDN	1 (2.2%)	0 (0.0%)	0.54
MMF/MPA + PDN	2 (4.3%)	1 (4.8%)	0.73
**A** **ntihumoral therapy N (%)**			
PPh + IVIG	--	12 (57.1%)	**--**
PPh + IVIG + rituximab	--	5 (23.8%)	**--**
PPh + IVIG + bortezomib	--	4 (19.0%)	**--**
Steroid bolus to treat concomitant TCMR	9 (19.6%)	10 (47.6%)	**0.04**
**Change of CNI after the studied biopsies N (%)**			
Increasing dose of CyA or Tac	10 (21.7%)	5(23.8%)	0.27
Reducing dose of CyA or Tac	9 (19.6%)	3 (14.3%)	0.36
Withdrawal of CyA or Tac	3 (6.5%)	1 (4.8%)	0.87
Switch between CyA and Tacrolimus	7 (15.2%)	7 (33.3%)	0.40
Switch CNI to mTor inhibitor	0 (0.0%)	0 (0.0%)	0.10
Switch CNI to Belatacept	5 (10.9%)	1 (4.8%)	0.56
No change	12 (26.1%)	4 (19.0%)	0.32
**Presence of advere events in the 6 months post studied biopsy**			
Urinary tract infection N (%)	3 (6.5%)	3 (14.3%)	0.45
Respiratory tract infection N (%)	1 (2.2%)	0 (0.0%)	0.44
CMV infectious colitis N (%)	0 (0.0%)	2 (9.5%)	**0.02**
Polyoma virus nephropathy N (%)	2 (4.4%)	0(0.0%)	0.09
Pancytopenia N (%)	3 (6.5%)	3(14.3%)	0.33
Overall	9 (19.6%)	8 (38.1%)	**0.04**
**The level of HbA1c and blood pressure at studied biopsy**			
HbA1c level at studied biopsy (% means ± SD)	5.6 ± 0.7	5.2 ± 0.4	**0.02**
SBP at studied biopsy (mmHg mean ± SD)	141.3 ± 19.5	129.4 ± 16.6	**0.02**
DBP at studied biopsy (mmHg mean ± SD)	83.4 ± 10.5	80.1 ± 14.1	0.29
**Antihypertensive therapy at studied biopsy**			
ACEi N (%)	24 (51.7%)	11 (52.4%)	0.71
ARB N (%)	16 (34.0%)	11 (52.4%)	0.14
CCB N (%)	27 (57.4%)	13 (61.9%)	0.87
Beta-blocker N (%)	7 (14.9%)	2 (9.5%)	0.64

cAMRwo: chronic antibody mediated rejection without treatment; cAMR-AHT: chronic antibody mediated rejection with antihumoral treatment; IQR: interquartile range; BMI: body mass index; CIT: cold ischemic time; PRA: panel reactive antibody; HLA: human leukocyte antigen; DSA: donor-specific anti-HLA antibodies; MMF: mycophenolate mofetil; MPA: mycophenolic acid; Tac: Tacrolimus; CyA: Cyclosporin A; PDN: Prednisolon; Pulse of Methylprednisolon: 500 mg/day for 3 days; Switch of calcineurin inhibitor (CNI): between Cyclosporin and Tacrolimus or initiating Rapamycin after suspending CNI; PPh: plasmapheresis; IVIG (Intravenous immunoglobulin): 1.5 g/kg; rituximab (anti-CD20 globulin): 375 mg/m^2^; repeated every 3 weeks for three rounds; bortezomib (therapeutic proteasome inhibitor): 1.3 mg/m^2^, administered by intravenous bolus on days 1, 4, 8 and 11 of a 21-day cycle; CMV: cytomegalovirus; HbA1c: hemoglobin A1c; SBP: systolic blood pressure; DBP: diastolic blood pressure; ACEi: angiotensin converting enzyme inhibitors; ARB: angiotensin receptor blocker; CCB: calcium channel blocker.

**Table 2 jcm-11-00199-t002:** Comparison of DSA, estimated glomerular filtration rate (eGFR) and proteinuria between groups.

	cAMRwo (*n* = 46)	cAMR-AHT (*n* = 21)	*p*-Value
**The values of MFI at and post the studied biopsies (median IQR)**			
MFI_max at biopsy, median (IQR)	7500 (528–23,336)	8293 (48–16,275)	0.98
MFI_max at 6 months post biopsy, median (IQR)	6234 (2018–18,209)	4470 (369–12,811)	0.61
MFI_max at 1 year post biopsy, median (IQR)	6368 (593–21,934)	4920 (43–16,817)	0.50
**The values of eGFR at and post the studied biopsies (mL/min median IQR)**			
eGFR value at 6 months before studied biopsy	35.3 (12.0–86.0)	40.1 (10.5–88.6)	**0.04**
eGFR value at studied biopsy	28.5 (5.4–67.9)	26.4 (10.0–52.0)	0.60
eGFR value at 6 months post studied biopsy	26.7 (5.0–89.0)	21.6 (5.0–52.0)	0.30
**The decline in eGFR values at and post the studied biopsies (mL/min median IQR)**			
∆ eGFR value at studied biopsy	−6.6 (−6.0–30.0)	−13.1 (−1.0–60.6)	**0.04**
∆ eGFR value at 6 months post studied biopsy	−2.6 (−31.0–25.0)	−4.9 (−18.8–7.0)	0.61
**The variation in the proteinuria at and post the studied biopsies (mg/day median IQR)**			
PU value at 6 months before studied biopsy	991(59–5155)	653 (45–2613)	0.09
PU value at studied biopsy	918 (48–11,579)	969 (143–5812)	0.78
PU value at 6 months post studied biopsy	665(89–6989)	1114 (208–3732)	0.57
**The decline in proteinuria at and post the studied biopsies (mg/day median IQR)**			
∆ PU value at studied biopsy	−21 (−3744–7465)	304 (−950–5588)	0.11
∆ PU value at 6 months post studied biopsy	89 (−8272–4318)	27 (−3145–1195)	0.47

MFI: mean fluorescent intensity; eGFR: estimated glomerular filtration rate; PU: daily urine protein excretion.

**Table 3 jcm-11-00199-t003:** Comparison of pathological features between groups.

	cAMRwo (*n* = 46)	cAMR-AHT (*n* = 21)	*p*-Value
**(A)** **Histological diagnosis in for-cause allograft biopsies before studied biopsies**			
Episode of TCMRi > = 1, N (%)	13 (28.3%)	8 (38.1%)	0.09
Episode of TCMRv > = 1, N (%)	4 (8.7%)	0(0.0%)	0.38
Episode of active AMR > = 1, N (%)	2 (4.3%)	5 (23.9%)	0.11
Episode of ATI > = 1, N (%)	11 (23.9%)	3 (14.3%)	0.49
**(B)** **Histological diagnosis in for-cause allograft biopsies after studied biopsies**			
Episode of cAAMR > = 1, N (%)	9 (19.6%)	4 (19.0%)	0.81
Episode of cAMR > = 1, N (%)	12 (26.1%)	9 (42.9%)	**0.03**
**(C)** **Histological scores of Banff lesions in the studied biopsy (scores mean ± SD)**			
g (0–3)	0.1 ± 0.3	0.4 ± 0.5	**0.01**
ci (0–3)	1.0 ± 0.9	0.5 ± 0.8	**0.04**
ct (0–3)	1.0 ± 0.9	0.5 ± 0.8	**0.04**
mm (0–3)	0.7 ± 0.9	0.3 ± 0.6	0.06
ptc (0–3)	0.1 ± 0.3	0.1 ± 0.2	0.08
ah (0–3)	2.5 ± 0.9	2.4 ± 0.8	0.12
cv (0–3)	1.7 ± 1.1	1.4 ± 1.0	0.17
v (0–3)	0.1 ± 0.3	0.1 ± 0.3	0.17
i (0–3)	0.7 ± 1.0	0.9 ± 1.0	0.27
cg (0–3)	2.1 ± 0.8	2.4 ± 0.7	0.33
t (0–3)	0.3 ± 0.7	0.5 ± 0.6	0.38
C4d (0–3)	0.0 ± 0.0	0.0 ± 0.0	0.99
Concomitant TCMR N (%)	9 (19.6%)	10 (47.6%)	**0.04**

TCMRi: acute T-cell-mediated interstitial rejection that includes Borderline changes, TCMR Ia and TCMR Ib; TCMRv: acute T-cell-mediated vascular rejection that includes TCMR IIa, IIb and III; ATI: acute tubular injury; Banff scored lesions: glomerulitis (g); peritubular capillaritis (ptc); transplant glomerulopathy (cg); intimal arteritis (v); interstitial inflammation (i); tubulitis (t); mesangial matrix increase (mm); vascular intimal fibrosis (cv); arteriolar hyaline thickening (ah); interstitial fibrosis (ci) and tubular atrophy (ct).

**Table 4 jcm-11-00199-t004:** Binary logistic-regression analysis of clinical factors associated with 5-year outcome after diagnosis of cAMR.

	Overall Graft Loss				Patient Death				Death-Censored Graft Loss			
Clinical factors	OR	95% CI		*p*-value	OR	95% CI		*p*-value	OR	95% CI		*p*-value
eGFR above median at biopsy	0.4	0.1	1.5	0.18	0.4	0.1	1.6	0.20	0.4	0.1	1.6	0.19
PU above median at biopsy	2.2	0.6	8.5	0.24	2.3	0.6	8.9	0.23	2.2	0.6	8.5	0.18
Receiving antihumoral therapy	2.5	0.6	10.0	0.21	1.9	0.5	7.8	0.38	2.6	0.6	10.7	0.18

eGFR: estimated glomerular filtration rate; PU: Proteinuria; OR: odds ratio; CI: conference intervals for odds ratio.

## Data Availability

The data presented in this study are available on request from the corresponding author. The data are not publicly available due to data protection and ethical restrictions.

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
