# Peer review of "Poor Long-Term Renal Allograft Survival in Patients with Chronic Antibody-Mediated Rejection, Irrespective of Treatment—A Single Center Retrospective Study"

_jcm, 2021, doi:10.3390/jcm11010199_

Round 1

Reviewer 1 Report

Chronic antibody-mediated rejection is a clinically
significant problem in renal transplant recipients.
Diagnosis of this type of rejection worsens the survival
of the transplanted kidney, significantly.
However, there are no clear rules
for the treatment of this type of rejection
in renal transplants recipients.
The authors tried fo find the optimal way
of dealing with chronic antibody-mediated rejection. The results of the study showed that the utility
of antihumoral treatment had no major effect
on the fluctuating course of DSA or improving the graft
Furthermore it created infectious complications for the patients.
The method of preparation and presentation of the described
study prove the authors' extensive experience in dealing with
kidney recipients. Moreover, the discussion reflects
the wide knowledge of the authors regarding discussed clinical problem.

Editorial notes - it is necessary to correct the font in lines 25, 35 and 351 - on line 24 in the abstract, there is an acronym ATH,
and it should be AHT as in the rest of the manuscript. - in Table 1, in the section entitled Clinical characteristics,
line 2 has been shifted one line.

Author Response

point 1:  it is necessary to correct the font in lines 25, 35 and 351 - on line 24 in the abstract, there is an acronym ATH

Response 1: the font of the words has been corrected.

point 2: on line 24 in the abstract, there is an acronym ATH

Response 2: sorry for the confusion, ATH was corrected to “antihumoral therapy”

point 3:  in Table 1, in the section entitled Clinical characteristics, line 2 has been shifted one line

Response 3: the shifted one line has been deleted and additional edits have been made in order to improve legibility

Reviewer 2 Report

The title of the article is "Poor long-term renal allograft survival in patients with chronic antibody-mediated rejection, irrespective of treatment - a single center retrospective study".
The authors conducted a retrospective cohort of 67 patients with chronic antibody-mediated rejection at the transplant centre of Charité Campus Mitte and Charité Virchow Klinikum. This study aimed to investigate long-term outcomes with a focus on the efficacy and safety of currently available agents for treatment of pure cAMR. You did a great job. However, some of main important issues need to be verified to improve your work as following.

I have the following comments,

  1. How was sample size determined? The main concern is the small sample sizes of each group of participants, which may have rendered it insufficiently powered to compare outcomes by using univariate and multivariate analysis. There is no discussion on power calculation in the statistical methods section, so the reader is left unsure whether the number of participants was enough to achieve power to detect differences at z% statistical power. This may pose the risk of committing type II errors. The results with too low statistical power will lead to weak conclusions about the meaning of the result.
  2. Include full details of how the authors handled missing data and outliers in the ‘Methods’ section.
  3. How did the authors assess the baseline proportionality of the Cox PH model? Please provide statistic test and visualization for confirm Cox PH assumption in supplementary explicit.
  4. In Table 2, if the patients had multiple laboratory data at admission, which one was included in analysis?
  5. In Table 4, the Cox proportional hazard model, if the prediction is their objective the model assumptions as well as model performance, a test for the interaction between variables, multicollinearity and goodness-of-fit analysis should be performed and show in the results or supplementary.

Author Response

point 1: How was sample size determined?

Response

As outlined in the revised version of the manuscript (line 76ff), this retrospective study focuses on all patients in our database, who meet the diagnostic criteria of cAMR according to Banff report 2017 and were followed in our database. 

We reviewed all cases who presented with the first episode of biopsy-proven TG lesion. In order to have a homogeneous and well-defined study population, we used a restricted cAMR definition, which was defined by the conservative threshold of cg>=1 and DSA positivity without C4d deposition or obvious microcirculation inflammation. Thus, all cases were excluded, if there was no HLA-examination for DSA at biopsy or if the biopsy showed only minimal C4d deposition or moderate/severe microcirculation inflammation or loss of follow-up shortly after studied biopsy. 

Although sample size is not large, this is one of the largest studies in the field, and currently the first study employing a strict definition of the population according to Banff 2017. The sample size of our study is similar or even larger compared to many other studies or clinical trials for the treatment of chronic AMR (see references 11, 19, 21, 25, 26, 27, 38).

Point 2: Include full details of how the authors handled missing data and outliers in the ‘Methods’ section.

Response

In the revised version we now have addressed this important point and have clarified how we handled missing data (see line 80 ff). As pointed out in figure 1, we excluded patients without adequate follow-up and included all data from our large and comprehensive transplant database, where we started in 1999 to prospectively collect all relevant data from our renal allograft recipients (see reference 13). This database captures all clinical data, medication and lab data from all outpatient visits and hospitalizations.

The table below outlines the proportion of missing data for key variables, which were always below 6%. In addition, we now describe in the revised version, how we handled missing data or selected data in case of several values within a time period (see line 84ff).

Table: completeness of key variables

% Missing

Variable

0%

Age, Gender

3.0%

Cold ischemic time (h)

4.5%

DGF(% yes)

1.5%

Primary Disease IgA, FSGS, Any GN, Hypertension, Diabetes

1.5%

Baseline HB1Ac (% yes)

1.5 %

PRA max (%)

1.5%

PRA at transplantation (%)

1.5%

BMI at Transplant

0%

Number of Previous Kidney Transplants

0%

Donor Age, Living,  Gender

1.5%

HLA MM (mean ± sd)

6.0%

Years on Dialysis, Preemptive

3.0%

FK, CyA, MPA, AZA, mTOR, Steroids for immunosuppression

6.0%

Proteinuria at biopsy

1.5%

eGFR at biopsy

6.0%

 SBP,  DBP

0%

Banff scored lesions (0-3)

6.0%

DSA-MFI

Point 4: in Table 2, if the patients had multiple laboratory data at admission, which one was included in analysis?

Response:

As pointed out in the revised manuscript (line 79ff), we always included only the routine measurements at- or nearest to the scheduled visit. Because data were collected during clinical routine, we did not have multiple values on the same day.

Point 3 and 5: How did the authors assess the baseline proportionality of the Cox PH model? Please provide statistic tests and visualization for confirming Cox PH assumption in supplementary explicit. In Table 4, the Cox proportional hazard model, if the prediction is their objective the model assumptions as well as model performance, a test for the interaction between variables, multicollinearity and goodness-of-fit analysis should be performed and show in the results or supplementary

Response:

In order to address the valid concern of the reviewer we thoroughly discussed the statistics section with a statistician. We agree that the low number of events per group limits the applicability of the Cox PH model according to the current recommendations like TRIOD statement. We also recognize the fact that histological and transplant characteristics are not continuous longitudinal data, which further limits the applicability of the Cox model.

After consultation with the statistician, and as suggested by the reviewer, we performed a visualization of the histological criteria using Kaplan-Meier graphs after dividing the Banff lesions scores into mild and severe. Because we did not observe statistical differences between mild and severe Banff lesion scores, we did not select histological lesions scores for further multivariable analysis (see Supplementary Figure 2 and Table 4).

In order to assess the impact of three selected clinical variables on 5-year outcome we now use a binary logistic regression analysis to estimate the effects eGFR, proteinuria and antihumoral treatment on the long-term outcome as described in the revised methods and results section (see line 166ff and 266ff). We also changed the discussion accordingly (line 395ff) .

Round 2

Reviewer 2 Report

Table 4: Binary logistic-regression analysis part, HR (hazard ratios) are analytic outcomes by using cox proportional hazard model, please change outcomes from HR (hazard ratios) to Odds Ratios.

Author Response

Table 4: Binary logistic-regression analysis part, HR (hazard ratios) are analytic outcomes by using cox proportional hazard model, please change outcomes from HR (hazard ratios) to Odds Ratios

response: 

we changed the "HR" to "OR" in Table 4 and note.